# Factors Influencing the Continued Intent to Use Virtual Interactive Platforms in Korean Small- and Medium-Sized Enterprises for Remote and Hybrid Work

**Hang To Diem Tran** [1] and **Minsook Kim** [2,*]

1   Graduate School of International Commerce, Keimyung University, Daegu 42601, Republic of Korea; cba143@naver.com
2   Department of Economics and Commerce, Keimyung University, Daegu 42601, Republic of Korea
*   Correspondence: kms@gw.kmu.ac.kr; Tel.: +82-53-580-5383

**Abstract:** Virtual interactive platforms have gained popularity in remote and hybrid work settings. However, limited research exists regarding factors that explain employees' continued use of these platforms, particularly in small and medium-sized enterprises (SMEs). This study aims to introduce a comprehensive research model that elucidates the relationships among platforms' technological characteristics, individual motivations, satisfaction, and behavioral intentions in the context of virtual interactive platforms. The authors conducted an empirical study involving 353 employees from SMEs in Daegu, South Korea, who utilized virtual interactive platforms for work. The study identified the platform features that enhance users' intention to continue using the service, and examined the impact of different motivations on users' behavioral intentions. The findings revealed that while telepresence and interactivity significantly influenced user satisfaction, they did not directly affect continued use intentions. User satisfaction and extrinsic motivation were positively associated with users' intention to continue using the platform. Among the identified motivations, extrinsic motivation had the strongest impact on continued use intention, whereas intrinsic motivation had the weakest impact. This study has important implications for managers aiming to optimize the benefits of virtual interactive platforms in hybrid work environments. Additionally, it offers insights for platform providers seeking to enhance their services. By understanding the factors that drive user satisfaction and continued use intention, organizations can develop strategies to maximize the sustainability advantages of virtual interactive platforms.

**Keywords:** virtual interactive platforms; sustainable hybrid work; organismic integration theory; user satisfaction; continued use intention

## 1. Introduction

The practice of remote and hybrid work, as a viable alternative to traditional office businesses, has been steadily growing among the workforce, a shift that has recently been made possible by innovations in virtual interactive platforms (VIPs) such as Zoom, Microsoft Teams, and Metaverse-based oVice. These virtual technology-based platforms enable organizations to develop a new hybrid work model in which employees usually enjoy greater autonomy and a better work-life balance by combining in-office and remote work [1]. Virtual interactive platform was initially adopted in the business sector to facilitate professional interactions between firms and distributed work [2], and then were extended into telehealth [3], distance education [4], and more recently, individual use, for example, in long-distance relations [5,6]. Meanwhile, restrictions imposed to counteract the recent COVID-19 pandemic have forced an unprecedented acceleration in the adoption of remote work globally. Several big technology companies such as Meta, Microsoft, Apple, and Nvidia are developing technologies that enable users to interact physically with virtual content in a three-dimensional space; however, these companies are not well established [7].

Therefore, virtual interactive platforms are currently the best choice for small- and medium-sized entrepreneurs (SMEs), who are unable to create their own virtual collaborative systems, to ensure a hybrid work model. Many institutions have chosen to utilize virtual interactive platforms for diverse purposes, with 85% of organizations using more than one meeting platform, and 62% of videoconferencing companies using more than three different platforms or forms of software [8].

Usage of virtual interactive services has been shown to impact the sustainability of hybrid work arrangements. Gibaldi and McCreedy [9] proposed that working via virtual support tools may promote productivity and connectedness among employees and managers during a certain period. Martin et al. [10] supported these findings by clarifying that digital collaboration and communication tools have a positive impact on the productivity and job satisfaction of teleworkers. The authors argue that digital collaboration tools can help to build trust and enhance communication among team members, which can lead to better decision-making and more effective collaboration. Seeber and Erhardt [11] confirmed that job satisfaction is influenced by the frequency with which information employees use virtual work tools. Based on the background of the literature, we can conclude that the effective use of virtual interactive services can support sustainable hybrid work by improving communication and collaboration among remote and in-person workers, and enhancing job performance and job satisfaction. Hence, it is essential to explore which factors affect workers' willingness to continue using a certain platform.

With an increasing number of platforms being established and introduced, users now have different choices on novel alternatives that utilize virtual technologies [7]. For example, oVice has recently offered 2D virtual spaces, wherein the user interacts as an avatar and connects with others around them, in a manner similar to an in-person interaction. To ensure work performance and productivity among hybrid teleworkers, managers need to consider (1) "What specific technical features drive employees to continue using a specific type of virtual interactive platform?", and find out the most sustainable and appropriate platform for their workplace. Previous studies in this field have primarily investigated the theoretical and practical impacts of virtual interactive platforms on meetings in the medical and academic communities [12–14], and limited research has been conducted to explore employees' attitudes and responses in SMEs' virtual workplaces is limited. Thus, we selected employees in SMEs as the research subject, in order to understand their intentional behavior in this specific setting.

In addition, a systematic literature review implemented by Gilal et al. [15] between 1991 and 2020 revealed that organismic integration theory (OIT) is a notable theory of human motivation that has significant interaction with the marketing environment to achieve positive promoting results, and can be used to analyze consumer loyalty [16] and purchase intention [17,18]; however, this element has been surprisingly ignored in the marketing literature, particularly in the area of virtual technologies. This leads to the second vital question: (2) How do different types of motivation influence users to continue using virtual interactive platforms?

To determine the answers to these questions, this study aims to take the first step in advancing the body of knowledge on this subject by presenting an integrative virtual technology usage model based on the concepts of technology features (interactivity and telepresence), motivation theory (OIT), user satisfaction, and continued use intention. The research findings provide comprehensive insights into the attitudes and behavioral intentions toward virtual interactive platforms of employees in hybrid positions toward virtual interactive platforms, helping managers and leaders of organizations choose suitable services that enhance collaboration and productivity among workers, and that even reduce environmental impact [19,20]. Additionally, this paper is also expected to form a valuable reference material for platform developers and operators who wish to better grasp the human motivational factors, user satisfaction factors, and technical factors that contribute to their continued intention to use these platforms.

## 2. Literature Review

### 2.1. Technological Features of Virtual Interactive Platforms

Virtual interactive platforms are services that help people in different locations communicate with each other using various devices such as computers, tablets, and mobile devices [21]. Following technological development, the platform system has grown to include numerous features (e.g., high-quality visuals, audio, and eye contact) to suit user needs in a virtual meeting. Zoom, Microsoft Teams, Cisco Webex, and oVice are examples of these services. Although virtual interactive platforms have various technological features, telepresence (TEL) and interactivity (INT) are the two most significant factors perceived by users when using virtual platforms to connect with others and implement tasks in a virtual environment [21–23].

The concept of 'telepresence' was first introduced by Minsky [24], who defined it as the degree to which the attendees of a remote meeting witness and experience occurrences in a different location through a feedback system utilizing virtual technology. Subsequently, many scholars have emphasized the meaning of telepresence as "a sense of being physically present in an unreal, imagined environment with visual and auditory" [25] or force signals generated by a mediated medium, such as virtual platforms, TV, or other media [26,27]. In virtual meeting systems, the environment mediated via platforms is expected to erase the boundaries between the physical and remote environments. Service creators and providers continuously attempt to make technological improvements, such as capturing and rendering a 3D audio-visual likeness of a remote person, and creating comfortable displays with a wider field of view or stereopsis. Therefore, this study thus defines TEL as a measure of the technical attributes of virtual interactive platforms that enable users to perceive a sense of visual vividness while communicating through a platform interface from a distance.

Despite the diversity in definitions of 'interactivity', there are two basic approach perspectives [21]. The first perspective describes it as a set of processes that occur in communication that might be affected by the exchange of opinions or oral content between people. Therefore, both computer-mediated and in-person communication can enable high interactivity [28]. Meanwhile, the second perspective, suggested by Steuer [25], defines interactivity as "a degree to which users of a medium can influence the form or content of the mediated environment." Interactivity can be separated into three dimensions: (1) Speed, which indicates the speed at which input could be assimilated into the mediated medium; (2) Range, which describes the number of action options available at any particular time; and (3) Mapping, which refers to a system's capacity to map its controls naturally and predictably with shifts in the mediated environment. The goal of virtual interactive platforms is to serve as a communication tool for the user engagement, regardless of location, time, or context. Therefore, virtual interactive platforms must be designed to satisfy users desire to communicate effortlessly and remain in control of technical issues in various environments. Based on the background concepts in Steuer's research [25], this study computes a measure of interactivity as an additive combination of a system's speed, range, and mapping. This involves measuring how quickly (speed), how much (range), and how naturally (mapping) users can control communication using the tools provided in the platforms.

### 2.2. Organismic Integration Theory

Organismic integration theory (OIT) is one of the sub-theories of self-determination Theory (SDT) that explains the process of human motivation in social contexts [29]. The theory was first introduced by Deci and Ryan [30] to provide a framework for classifying different types of extrinsic motivation, including (1) intrinsic motivation, which is fully integrated with one's sense of self; (2) identified motivation, which arises from personal values and goals; (3) introjected motivation, which is driven by internal pressures, such as guilt or shame; and (4) external motivation, which is based on external rewards or punishments [15,16]. OIT proposes that individuals have varying degrees of self-determination or

autonomy in their actions, which can influence their motivation and engagement [30]. OIT also highlights the importance of internalization, which refers to the process of integrating external regulations or values into one's sense of self. It suggests that individuals are more likely to engage in sustained and meaningful behavior when they have internalized the values and regulations associated with their actions [31].

In this study, intrinsic motivation (INTR) is defined as user engagement in a specific virtual interactive platform for personal feelings of pleasure and interest that are directly derived from participation in the platform. Identified motivation (IDE) represents an individual recognition of the importance of a platform that aligns with their personal goals. Introjected motivation (INJ) presents when users engage in platform-using behavior to promote ego enhancement and obtain approval or commendation from others (e.g., work partners, colleagues, etc.), or to avoid feelings of shame, anxiety, or guilt. Finally, external motivation (EXT) includes platform use to satisfy external demands, avoid punishment, and comply with social pressures.

Building upon the existing literature, this study refines the conceptualization of each factor and presents their operational definitions, which are outlined in Table 1.

**Table 1.** Operational definition of each construct.

| Construct | Definition | Reference |
|---|---|---|
| Telepresence | The degree to which a user expects to perceive a sense of visual vividness while communicating through a platform interface from a distance. | [24–27] |
| Interactivity | The degree to which a user can control the speed, range, and mapping of communication using the provided tools. | [21,25] |
| Intrinsic Motivation | The degree of personal pleasure and interest that are directly derived from engagement with the platform. | [15,16,29,32,33] |
| Identified Motivation | The level of an individual's recognition of the importance of a platform that aligns with their personal goals. | [15,16,29,32,33] |
| Introjected Motivation | The motivation to use platforms to avoid guilt, gain approval, or meet standards set by their communities. | [15,16,29,32,33] |
| External Motivation | The motivation to use platforms to satisfy external demands, avoid punishment, and comply with social pressures. | [15,16,29,32,33] |
| User Satisfaction | The emotional state of users that can be gained from experiencing a specific technological application or online service. | [34–36] |
| Continued Use Intention | The office employees' intention to continue using the virtual interactive platforms for work. | [37–39] |

*2.3. User Satisfaction and Continued Use Intention*

Satisfaction refers to the measure of the level of pleasurable/the feeling of being fulfilled by a product or service after comparing the product's quality with the user's expectations [34]. User satisfaction, however, refers to the emotional state of users that can be gained from experiencing a specific technological application or online service [35,36]. The degree of satisfaction has an important impact on marketing strategies because it is significantly related to brand engagement [40,41], loyalty [40,42], and behavioral intention [43,44] in various contexts, including technology-based services such as virtual interactive platforms.

The term "behavioral intention" is first mentioned in the Theory of Reasoned Actions by Fishbein and Ajzen [37]. This theory explains that behavioral intentions can help to determine individual behavior, and are influenced by subjective norms and attitudes. Then, the concept of continued use intention was suggested by Davis [45] in the Technology Acceptance Model (TAM), which refers to the intention or likelihood of an individual to continue using a particular technology or system, or the likelihood of them continuing to use it. It is a concept commonly used in the field of information systems and technology acceptance research [46]. It aims to understand the factors that influence users' decisions to continue using a technology or to discontinue its use. Moreover, Baker and Crompton [38]

asserted that individuals' behaviors can be predicted based on their underlying intentions. Consequently, it is possible to estimate individuals' actual behaviors if it is feasible to precisely measure their intentions.

## 3. Hypothesis Development

### 3.1. Influence of Technical Features on Satisfaction and Continued Use Intention

A virtual platform is considered a tool for collaborating, progressing, and enabling people to stay connected in work communities [47]. Telepresence and interactivity are vital elements of a range of virtual technologies [43], especially virtual reality (VR), and are expected to be advanced in some novel versions of virtual interactive platforms. The current study is meaningful, as long as we clarify the connections between these factors and user satisfaction and continued usage intent in order to anticipate user reactions and behaviors.

Roy Dholakia and Zhao [48] identified that, in the context of retail websites, while telepresence has a direct impact on behavioral intentions, both interactivity and telepresence positively affect customer satisfaction, which may encourage the behavioral intention of customers. The important role of telepresence in promoting Chinese user satisfaction and their continued intent to use micro-blogging services featured in the research of Zhao and Lu [49]. The authors proposed that by improving this technical feature, service providers can satisfy potential users and enhance their competitive advantages. Al-Geitany et al. [50] revealed a positive relationship between perceived interactivity and user behavioral intentions in the context of virtual conferences in the post-COVID-19 era. Meanwhile, this feature also was considered by Bacolod et al. [51] as a fundamental characteristic of video-conferencing, which offers more learning benefits to students and encourages them to use this service rather than other alternatives.

According to previous studies [48–51], the technological features of each platform may be regarded as factors that promote the user's satisfaction and continued use intention regarding a particular service, as stated in the following hypotheses:

**H1.** *Interactivity positively impacts satisfaction;*

**H2.** *Telepresence positively impacts satisfaction;*

**H3.** *Interactivity positively impacts continued use intention; and*

**H4.** *Telepresence positively impacts continued use intention.*

### 3.2. Relationships between Satisfaction and Organismic Integration Theory

Prior studies have proven that the satisfaction of users with a product or service is related to their motivation as a critical antecedent. In other words, when satisfaction is achieved, users are significantly motivated by their inherent interest in the platform (intrinsic motivation), awareness of its importance (identified motivation), feelings of worth or avoidance of anxiety (introjected motivation), compliance with rules, and avoidance of punishment (external motivation). Lin et al. [16] integrated expectation–confirmation theory and self-determination theory to construct a new research framework, which proposed that satisfaction is an important predictor of self-determined motivation. The studies of Rahi et al. [32,33] on the behavior of Internet banking users proved that satisfaction has a positive relationship with four regulations (intrinsic motivation, introjected regulation, identified regulation, and external regulation) of extrinsic motivation. Standage et al. [52] found that satisfaction may predict intrinsic motivation, which in turn created the outcomes of school physical education.

When using virtual interactive platforms for work, users must have their own purpose (e.g., presentation, business meetings, cooperation tasks, etc.), and will then reuse the same platform for similar tasks and goals. Thus, if users are satisfied with a certain platform, they tend to reuse it. Therefore, we can underpin the relationship between the satisfaction and motivation of virtual interactive platform users with the following hypotheses:

**H5.** *Satisfaction positively impacts intrinsic motivation;*

**H6.** *Satisfaction positively impacts identified motivation;*

**H7.** *Satisfaction positively impacts introjected motivation; and*

**H8.** *Satisfaction positively impacts external motivation.*

### 3.3. Relationships between Organismic Integration Theory and Continued Use Intention

In virtual interactive platform use, individuals may decide to utilize a platform to achieve their objective of connecting with others and to satisfy external demands such as company and staff needs. The use of virtual interactive platforms in hybrid work arrangements has been found to positively impact productivity and connectedness [9], improve communication, and enhance decision making and collaboration among team members [10]. Therefore, understanding which factors promote employees to continue using a particular platform may give managers a chance to promote sustainable development within their organization.

Several scholars have confirmed that extrinsic motivations are vital for verifying users' continued use intentions. Deci and Ryan [30,31] indicated that extrinsic motivational elements enhance users' confidence in their usage of services. More recently, in a study on continuing use of e-learning technology by teachers, the authors hypothesized the relationship between intrinsic motivation and user intention, then concluded that intrinsic motivation has a positive impact on the degree of teachers' continued use intentions [53]. Rahi et al. [54] also determined the significant impacts of extrinsic motivation regulations on users' intention to continue using of e-banking during the recent pandemic. Accordingly, this study derived the association between the OIT and platform users' continued use intention as follows:

**H9.** *Intrinsic motivation positively impacts continued use intention;*

**H10.** *Identified motivation positively impacts continued use intention;*

**H11.** *Introjected motivation positively impacts continued use intention; and*

**H12.** *External motivation positively impacts continued use intention.*

### 3.4. Impact of Satisfaction on Continued Use Intention

Within the framework of the expectation confirmation theory, Lin et al. [39] investigated the intention to continue using a website and discovered that satisfaction was the most influential factor. The TAM model of Davis [45] also suggests that user satisfaction directly influences the intention to continue using a technology or platform. Furthermore, Awad et al. [44] confirmed that user satisfaction is significant in students' continued intention to use e-learning. Other studies have also provided evidence of a considerable positive impact of satisfaction on users' intentions to engage in technology-related behaviors [38,48,49].

Considering the existing features of platform services that empower users to regulate meeting quality and interact with others based on their individual objectives, this study posits a hypothesis stating that higher satisfaction with virtual interactive platforms leads to a stronger intention to continue using them. The hypothesis is formulated as follows:

**H13.** *Satisfaction positively impacts continued use intention.*

To present the hypotheses in a concise and visual manner, we have depicted the illustrative research model in Figure 1.

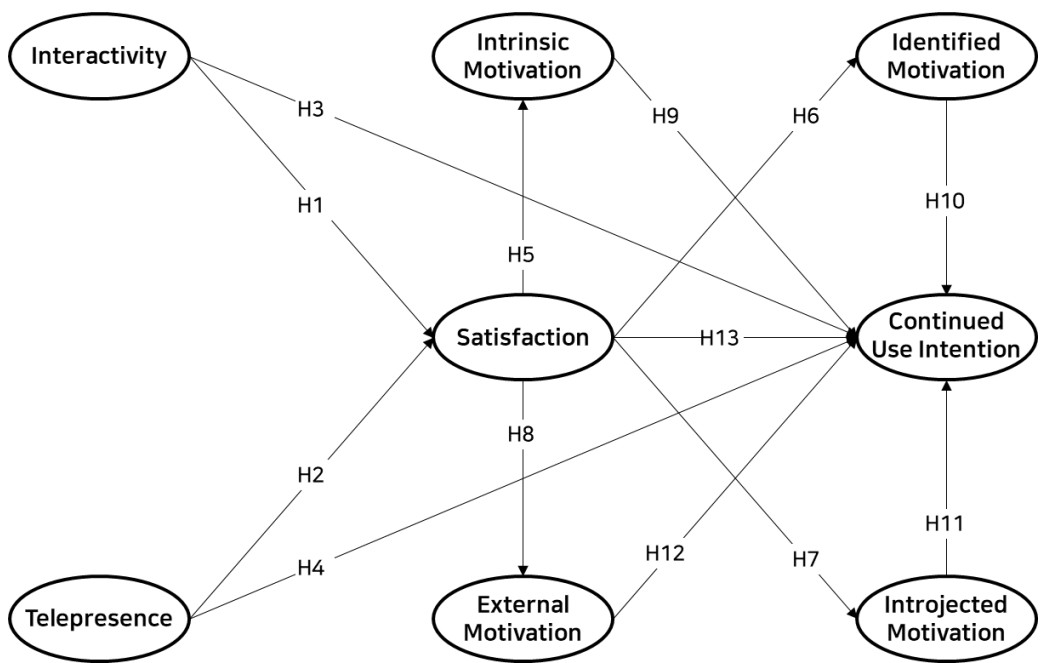

**Figure 1.** The research framework and hypotheses.

## 4. Materials and Methods

### 4.1. Participants and Procedure

This study, which constitutes part of a larger study targeting office employees of virtual interactive platform user communities, used a quantitative and cross-sectional approach. Data were collected in Daegu, Republic of Korea. A preliminary survey of 32 employees from a trading company and an academic institution was conducted to improve the questionnaire's validity and reliability. Johanson and Brooks [55] confirmed in their study that a reasonable minimum sample size for a pilot study is approximately 30 representative participants; thus, the sample size for the preliminary survey in this study is appropriate (N = 32). Based on the preliminary survey, we revised some of the contents of the questionnaire and then employed that edited questionnaire in the main survey. To ensure all participants were suitable for the research purpose, and that they perceived each specific service accurately, the questionnaire started with "Have you used virtual interactive platforms as an office employee?" and "What virtual interactive platform do you mainly use for work?".

To mitigate sampling bias, this study utilized two distinct approaches to collect data over a period of 40 days, from 20 January to 28 February 2022. The first approach involved an in-person survey employing the snowball sampling method. This method facilitated the collection of data from users who utilized platforms such as Zoom, Microsoft Teams, Webex, and others to communicate with colleagues and accomplish work-related tasks. More specifically, participants from trade companies (12 users) and design firms (3 users), and staff members of academic institutions (8 users) were invited to recommend other users in their communities to the survey. All participants volunteered to get involved in the research survey after receiving a clear explanation of the nature of the study from the researchers. Secondly, this study employed a random sampling technique to collect data. The online survey link, along with a comprehensive explanation of the research objectives, was shared on various online communication platforms that serve multiple purposes, including work, training, and job-seeking. This ensured a randomized selection of participants for data collection. We also ensured that all meaningful participant information was used and protected based on Sections 31 and 33 of the Statistics Act of Statistics Korea (KOSTAT) [56].

The demographic and general characteristics of the respondents were 186 women (52.7%), 162 men (45.9%), and 5 people who did not indicate their gender (1.4%). The age

group with the greatest response rates was 20–29 years (77.1%), followed by 30–39 years (18.4%), 40–49 years (2.8%), and 50–59 years old (1.7%), demonstrating that young adults in their 20s and 30s had the most experience using virtual interactive platforms at work. According to survey responses, the most popular services were Zoom (46.7%), Microsoft Teams (21.8%), oVice (19.3%), WebEx (9.9%), and others (Google Meet, NAVER WhaleOn, etc.) at 2.3%. Of those who responded to the survey, 96% had used the virtual platforms for between one and two years, 2.3% for more than two years, and 1.7% for less than a year. We have summarized the demographic analysis outcomes in Table 2.

**Table 2.** Demographic profiles of respondents.

| Measure (N = 353) | Items | Frequency | Percent (%) |
|---|---|---|---|
| Name of platforms | Zoom | 165 | 46.7 |
| | Microsoft Teams | 77 | 21.8 |
| | oVice | 68 | 19.3 |
| | Cisco WebEx | 35 | 9.9 |
| | Others | 8 | 2.3 |
| Gender | Male | 162 | 45.9 |
| | Female | 186 | 52.7 |
| | No response | 5 | 1.4 |
| Age groups | 20–29 | 272 | 77.1 |
| | 30–39 | 65 | 18.4 |
| | 40–49 | 10 | 2.8 |
| | 50–59 | 6 | 1.7 |
| Use period | Less than 1 year | 6 | 1.7 |
| | 1–2 years | 339 | 96.0 |
| | More than 2 years | 8 | 2.3 |

### 4.2. Methods Setting and Sample

By the end of the data collection period, 12 of the 365 returned survey responses were deemed unsuitable, leaving 353 for the final analysis. The number of responses was regarded as optimal if it was greater than 200, as proposed by Thompson [57]. Calculating an appropriate sample size is crucial for reducing sampling errors in the measurement model. Traditional sample size calculation rules, such as power, effect size, or minimal sample size, often overlook the specific characteristics of the study model. Therefore, it is imperative to consider the unique specifications of the study model when determining the sample size [58].

To establish an appropriate sample size for the model structure and detect the desired effect, this study employed the "A priori sample size calculator for structural equation models" developed by Dr. Daniel Soper [59]. The calculation considered several parameters: an expected effect size of 0.3 (medium), a desired statistical power level of 0.8, eight latent variables, 28 observable variables, and a significance level of 0.05 [60]. The calculated minimum recommended sample size was found to be N = 177. Given that the actual sample size for this study was N = 353, which exceeded the minimum recommended size, the results obtained from such a dataset can be deemed valid, reliable, and capable of enhancing the generalizability of the theoretical model.

### 4.3. Measures

The variables employed in this study consist of measurement items that have undergone validation for reliability and validity in prior research. Certain questionnaire items were modified to align with the context of virtual interactive platforms. Following the exploratory factor analysis, 4 items were eliminated based on the results of the pilot test involving 32 users. Additionally, item wordings were revised, taking into account feedback from experts and participants. Consequently, a final questionnaire scale comprising 28 items was established, encompassing 8 dimensions across all variables. Detailed measurement parameters can be found in Table 3 below.

**Table 3.** Measurement items.

| Construct | | Measurement Items | Source |
|---|---|---|---|
| Telepresence | TEL1 | I feel a sense of visual vividness as though I am meeting in person. | [21] |
| | TEL2 | I can easily recognize the response of participants due to the high video resolution. | |
| | TEL3 | I can see many participants in the meeting due to the adequate screen size. | |
| | TEL4 | I feel the composition and visual harmony of the system appear to be good. | |
| Interactivity | INT1 | I can quickly control how users interact or participate in a particular conversation. | [21] |
| | INT2 | I can instantly reject unwanted requests (e.g., recording, mute, etc.) | |
| | INT3 | The connection is stable and error-free during use. | |
| | INT4 | Sound and video are responsive. | |
| Intrinsic Motivation | INTR1 | I enjoy adopting the platform to improve my work. | [61–63] |
| | INTR2 | I find communicating through the platform a pleasurable activity. | |
| | INTR3 | I like using the platform to connect with others. | |
| Identified Motivation | IDE1 | This is how I can improve my work performance. | [61–63] |
| | IDE2 | This is how I communicate better with my colleagues. | |
| | IDE3 | I think it will help me in my job. | |
| Introjected Motivation | INJ1 | I would feel bad if I did not use it. | [61–63] |
| | INJ2 | I would feel ashamed if I did not use it. | |
| | INJ3 | I want my colleagues to think I am a good colleague. | |
| External Motivation | EXT1 | I get in trouble if I do not use it. | [61–63] |
| | EXT2 | This is what I am supposed to use. | |
| | EXT3 | It is requested by my organization. | |
| User Satisfaction | SAF1 | I am satisfied with my decision to use my virtual interactive platform. | [64] |
| | SAF2 | My choice to use this virtual interactive platform was a wise one. | |
| | SAF3 | I think I did the right thing by deciding to use my virtual interactive platform. | |
| | SAF4 | If I were to do it again, I would feel similarly about using my virtual interactive platform. | |
| Continued Use Intention | CUI1 | I intend to continue using my virtual interactive platform rather than discontinuing its use. | [21,65] |
| | CUI2 | I intend to increase the number of times I use my virtual interactive platform in the future. | |
| | CUI3 | I intend to keep using my virtual interactive platform as regularly as I do now. | |
| | CUI4 | I have no intention of trying another virtual interactive platform. | |

The antecedents comprised the variables of telepresence and interactivity, which were measured based on Tran's work [21]. Tran [21] developed reliable measurement scales to assess the interactivity and telepresence of virtual interactive platforms using a mixed-method approach, combining qualitative and quantitative research methods. Four OIT variables were rephrased based on this research context, following the scale in studies by Gegenfurtner and Quesada-Pallarès [61], James et al. [62], and Fellows [63].

Other variables included measures of user satisfaction toward virtual interactive platforms and continued use intention. User satisfaction was measured using four items based on Bhattacherjee [64], while continued use intention for a specific platform was assessed using four items adapted from Tran [21] and Shiau et al. [65]. A seven-point Likert scale ranging from "strongly disagree" (1) to "strongly agree" (7) was employed for each construct.

*4.4. Research Methodology*

This study used literature and empirical research methods to achieve the mentioned purposes. Regarding a conceptual approach, this study checked the factors and measures of individual intention, motivation, satisfaction, and features of virtual technology-based platforms from prior studies [12,14,21]. In terms of an empirical approach, we analyzed the collected data using structural equation modeling (SEM), which followed the two-step process recommended by Hair et al. [66]. More specifically, an exploratory factor analysis (EFA) was performed using the IBM SPSS 25.0 program to examine the structure of the predictor variables. We then used IBM AMOS 26.0 software to run a confirmatory factor analysis (CFA) on the measurement model and test constructed hypotheses using structural path analysis.

While EFA aids researchers in determining the optimal number of common factors and identifying which measured variables are valid indicators of the various latent dimensions [67], CFA allows researchers to assess the goodness-of-fit of their models and understand the significance and directionality of the relationships between variables [68]. This approach is also suitable for model conceptualization, recognition, and possible model

adjustment [66]. Since this paper aims to explore the relationships between several variables and test the underlying structure, it is appropriate to apply CFA to the statistical analysis.

Evaluation of the measurement model involved estimating the internal consistency of the constructs, including construct reliability, convergent and discriminant validity tests, and model-fit checks [66,67]. The Cronbach's alpha and composite reliability (CR) coefficients were used to evaluate construct dependability. The model fit was estimated using multiple indices (e.g., GFI, CFI, and TLI), as suggested by Hair et al. [66]. Convergent validity, which measures the extent to which items within a construct share a large portion of the variance [57], was evaluated using the average variance extracted (AVE) and factor loadings. Discriminant validity, which verifies the independence of each construct with respect to others [66], was determined by comparing the AVE to the maximum shared value (MSV) and the square root of the AVE to the inter-construct correlation.

## 5. Results

### 5.1. Validity and Reliability

Table 4 presents the results of the validity and reliability tests. Before examining the structural link between the variables, we conducted an EFA using the Varimax rotation method to determine whether the items accurately reflected their respective factors. The EFA resulted in an eight-factor solution that explained 72.270% of the total variance, with each factor having an eigenvalue greater than 1. The Kaiser Meyer Olkin was 0.904, which was greater than the acceptable range of 0.70, whereas the Bartlett sphericity test [$\chi^2$ (378) = 5075.702, $p < 0.001$] was statistically significant, implying that the data were appropriate for deeper analysis [69]. Each item in the eight categories has a factor loading greater than 0.50 (shown in Appendix A), which indicates conceptual validity.

**Table 4.** Confirmatory factors and reliability results.

| Items | Exploratory Factor Analysis | | | | Confirmatory Factor Analysis | | | Cronbach's Alpha |
|---|---|---|---|---|---|---|---|---|
| | Factor Loadings | Extraction | Eigenvalue | Cumulative % | CR | AVE | MSV | |
| TEL1 | 0.794 | 0.791 | | | | | | |
| TEL2 | 0.776 | 0.837 | | | | | | |
| TEL3 | 0.754 | 0.820 | 9.214 | 32.907 | 0.857 | 0.601 | 0.283 | 0.856 |
| TEL4 | 0.781 | 0.816 | | | | | | |
| INT1 | 0.768 | 0.776 | | | | | | |
| INT2 | 0.744 | 0.780 | | | | | | |
| INT3 | 0.766 | 0.815 | 2.442 | 41.629 | 0.833 | 0.557 | 0.269 | 0.833 |
| INT4 | 0.760 | 0.783 | | | | | | |
| INTR1 | 0.818 | 0.700 | | | | | | |
| INTR2 | 0.751 | 0.799 | 1.047 | 72.270 | 0.832 | 0.623 | 0.283 | 0.827 |
| INTR3 | 0.780 | 0.781 | | | | | | |
| IDE1 | 0.763 | 0.816 | | | | | | |
| IDE2 | 0.834 | 0.774 | 1.058 | 68.531 | 0.855 | 0.663 | 0.334 | 0.855 |
| IDE3 | 0.832 | 0.799 | | | | | | |
| INJ1 | 0.792 | 0.780 | | | | | | |
| INJ2 | 0.830 | 0.766 | 1.428 | 64.752 | 0.843 | 0.642 | 0.259 | 0.843 |
| INJ3 | 0.812 | 0.799 | | | | | | |
| EXT1 | 0.852 | 0.780 | | | | | | |
| EXT2 | 0.831 | 0.800 | 1.436 | 59.653 | 0.857 | 0.667 | 0.261 | 0.857 |
| EXT3 | 0.833 | 0.818 | | | | | | |
| SAF1 | 0.694 | 0.812 | | | | | | |
| SAF2 | 0.787 | 0.757 | | | | | | |
| SAF3 | 0.756 | 0.781 | 1.891 | 48.381 | 0.832 | 0.554 | 0.415 | 0.829 |
| SAF4 | 0.734 | 0.785 | | | | | | |

**Table 4.** *Cont.*

| Items | Exploratory Factor Analysis | | | | Confirmatory Factor Analysis | | | Cronbach's Alpha |
|---|---|---|---|---|---|---|---|---|
| | Factor Loadings | Extraction | Eigenvalue | Cumulative % | CR | AVE | MSV | |
| CUI1 | 0.723 | 0.759 | | | | | | |
| CUI2 | 0.725 | 0.768 | 1.721 | 54.526 | 0.844 | 0.579 | 0.415 | 0.837 |
| CUI3 | 0.762 | 0.808 | | | | | | |
| CUI4 | 0.601 | 0.835 | | | | | | |
| | KMO(Kaiser Meyer Olkin) | | | | | | 0.904 | |
| | Bartlett's test of sphericity | | | | Chi-Square | | 5075.702 | |
| | | | | | df (p) | | 378 (0.000) | |

Notes: AVE: average variance extracted, CR: composite reliability, MSV: maximum shared variance; criteria: CR > 0.7, AVE > 0.5, MSV < AVE.

According to Fornell and Larcker [70], a standardized coefficient of at least 0.50 is considered sufficient to establish the influence of a latent variable on an observed variable. For factor internal consistency, composite reliability (CR) was utilized, with a value higher than 0.70 indicating high reliability. The average variance extracted (AVE) should exceed 0.50. Furthermore, discriminant validity is achieved when the AVE surpasses the maximum shared variance (MSV) [66,71]. The CFA findings, as shown in Appendix C, revealed that each construct was distinct from the others, with elements primarily related to their own factors. Composite reliability was evidenced by Cronbach's alpha values and CR coefficients of 0.70 and above. The CR coefficients and alpha values were significantly higher than 0.80, indicating acceptable levels for all constructs. The AVE for every construct was above 0.50, presenting proof of convergent validity [66,70]. Meanwhile, the AVE was greater than the MSV, and the AVE's square root was greater than the inter-construct correlations, indicating discriminant validity, as shown in Table 5.

**Table 5.** Correlation and discriminant validity results.

| Factors | | 1 | 2 | 3 | 4 | 5 | 6 | 7 | 8 |
|---|---|---|---|---|---|---|---|---|---|
| 1 | INT | **0.746** | | | | | | | |
| 2 | TEL | 0.408 ** | **0.775** | | | | | | |
| 3 | SAF | 0.422 ** | 0.371 ** | **0.744** | | | | | |
| 4 | INTR | 0.432 ** | 0.463 ** | 0.291 ** | **0.789** | | | | |
| 5 | IDE | 0.325 ** | 0.323 ** | 0.374 ** | 0.418 ** | **0.814** | | | |
| 6 | INJ | 0.333 ** | 0.404 ** | 0.297 ** | 0.383 ** | 0.414 ** | **0.801** | | |
| 7 | EXT | 0.236 ** | 0.253 ** | 0.437 ** | 0.221 ** | 0.165 ** | 0.200 ** | **0.817** | |
| 8 | CUI | 0.444 ** | 0.441 ** | 0.551 ** | 0.460 ** | 0.495 ** | 0.421 ** | 0.423 ** | **0.761** |

Notes: *p*-value: ** <0.01; diagonal coefficients (in bold) reflect the square root of AVE.

### 5.2. Model Fit

The structural model was tested for goodness-of-fit before the hypothesis test. As shown in Table 6, the RMSEA was 0.023, GFI was 0.929, CFI was 0.988, TLI was 0.985, and NFI was 0.927, while the PCLOSE was higher than 0.05. When taken as a whole, these final indices prove the excellent fit of the proposed model to the data [66].

**Table 6.** Results of the model fitting.

| Index | CMIN ($\chi^2$) | CMIN ($\chi^2$)/df | RMSEA | PCLOSE | CFI | TLI | NFI | AGFI | GFI |
|---|---|---|---|---|---|---|---|---|---|
| Results | 382.119 (0.012) | 1.187 | 0.023 | 1.000 | 0.988 | 0.985 | 0.927 | 0.911 | 0.929 |
| Criteria | $p \geq 0.05$ | $1.0 \leq \chi^2/df \leq 2.0$ | ≤0.05 | ≥0.05 | ≥0.90 | ≥0.90 | ≥0.90 | ≥0.90 | ≥0.90 |
| Source | [72] | [73] | [74] | [75] | [73] | [75] | [75] | [76] | [76] |

### 5.3. Hypothesis Testing

The findings of the hypothesis testing are presented in detail in Table 7. To assess the impact of latent variables, regression weights are employed to determine the relative weights, while the standard error (S.E.) is utilized to evaluate parameter accuracy. Hypothesis acceptance is determined based on the significance level (*p*-value) associated with the critical ratio (C.R.) value. In a two-sided test, a C.R. exceeding 1.96 is considered significant at a *p*-value of 0.05, while a value surpassing 2.58 is deemed significant at 0.01.

**Table 7.** Hypothesis results.

| | Hypothesis | | | Path Coefficient | S.E. | C.R. | *p*-Value | Result |
|---|---|---|---|---|---|---|---|---|
| [H1] | INT | → | SAF | 0.397 | 0.069 | 5.737 | *** | Accepted |
| [H2] | TEL | → | SAF | 0.258 | 0.055 | 4.707 | *** | Accepted |
| [H3] | INT | → | CUI | 0.030 | 0.047 | 0.644 | 0.519 | **Rejected** |
| [H4] | TEL | → | CUI | 0.062 | 0.037 | 1.658 | 0.097 | **Rejected** |
| [H5] | SAF | → | INTR | 0.614 | 0.085 | 7.218 | *** | Accepted |
| [H6] | SAF | → | IDE | 0.680 | 0.085 | 8.019 | *** | Accepted |
| [H7] | SAF | → | INJ | 0.517 | 0.074 | 6.956 | *** | Accepted |
| [H8] | SAF | → | EXT | 0.497 | 0.062 | 8.061 | *** | Accepted |
| [H9] | INTR | → | CUI | 0.073 | 0.032 | 2.302 | 0.021 * | Accepted |
| [H10] | IDE | → | CUI | 0.136 | 0.034 | 3.957 | *** | Accepted |
| [H11] | INJ | → | CUI | 0.085 | 0.037 | 2.278 | 0.023 * | Accepted |
| [H12] | EXT | → | CUI | 0.129 | 0.046 | 2.803 | 0.005 ** | Accepted |
| [H13] | SAF | → | CUI | 0.262 | 0.078 | 3.374 | *** | Accepted |

Notes: $p$ = *** <0.001; ** <0.01; * <0.05; S.E.: standard error; C.R.: critical ratio.

According to the results, telepresence and interactivity were verified to be positively associated with user satisfaction, with β = 0.258 ($p < 0.001$) and β = 0.397 ($p < 0.001$), respectively, thus confirming H1 and H2. However, these two technical factors did not directly correlate with continued use intentions; therefore, H3 and H4 are rejected. Regarding user satisfaction and OIT constructs, the results indicate that user satisfaction had a strong positive influence on intrinsic motivation, with β = 0.614 ($p < 0.001$), and identified motivation, with β = 0.680 ($p < 0.001$), thereby confirming H5 and H6.

Similarly, H7 and H8 reveal that user satisfaction had an important impact in determining virtual interactive platform users' introjected motivation, with β = 0.517 ($p < 0.001$), and external motivation, at β = 0.497 ($p < 0.001$). Additionally, user satisfaction was positively associated with the continued use intention of users, with β = 0.262 ($p < 0.001$), thus confirming H13. Examination of the OIT-related path coefficients showed that all four dimensions of the OIT constructs were significantly associated with continued use intentions. Based on these results, H9, H10, H11, and H12 are all acceptable. Figure 2 below (or Appendix B) visually summarizes the results of the proposed hypotheses.

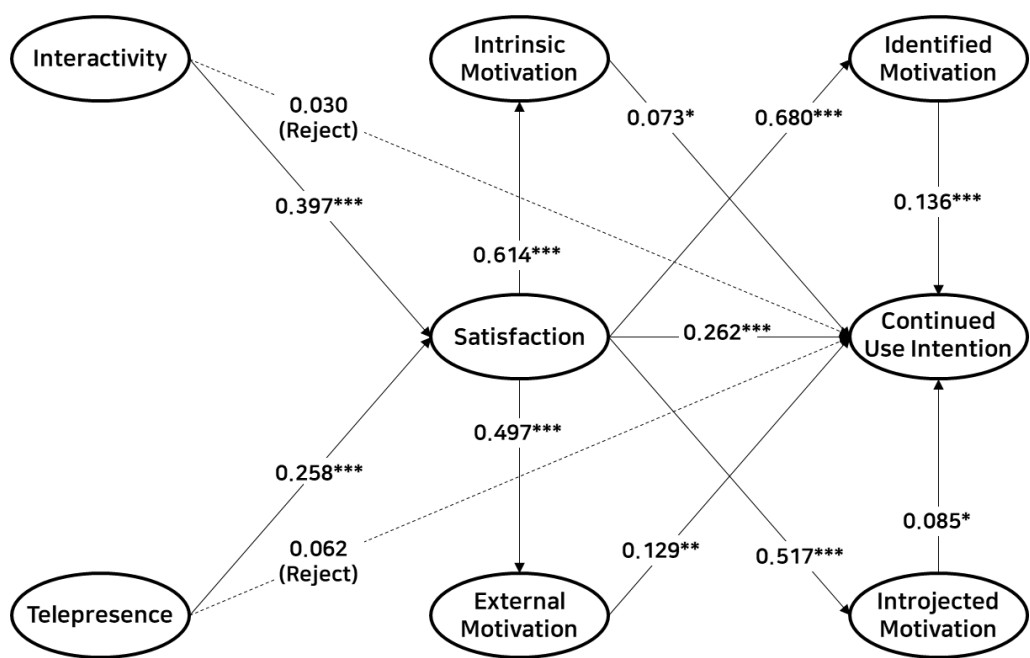

**Figure 2.** Hypothesis results summary. Notes: *** $p < 0.001$; ** $p < 0.01$; * $p < 0.05$.

## 6. Discussions

### 6.1. Results Summary

The purpose of this paper is to introduce an integrated research model that explains the complex relationships between platform technological characteristics, individual motivations, satisfaction, and behavioral intentions in the context of virtual interactive platforms. We aim to provide theoretical and practical implications for managers to maximize the sustainability benefits of virtual communication tools in hybrid work environments, as well as meaningful insights for platform developers to improve their services. This study addresses two crucial questions: (1) What technical features drive employees to continue using a specific virtual interactive platform? and (2) How do different types of motivation influence users to continue using virtual interactive platforms? The summarized results are as follows.

Regarding the first question, the study found that telepresence and interactivity do not directly impact users' intention to continue using virtual interactive platforms, despite their significant effect on user satisfaction. This means that the quality of the technology provided by virtual interactive platforms has no bearing on the decision of users to continue using them. These results contrast with previous findings by Roy Dholakia and Zhao [48], Zhao and Lu [49], Al-Geitany et al. [50], and Ying et al. [77], wherein telepresence and interactivity had positive effects on users' behavioral intentions. One possible explanation is that the impact of telepresence and interactivity on user intentions may be mediated by other factors. In this case, these factors include satisfaction and motivation regulations. However, the study of Wiardi et al. [78] on online learning platforms supports the present study's findings by demonstrating that technological quality had no direct effect on the intention to reuse platforms.

Next, we examined the association between user satisfaction, the four motivation types of OIT constructs, and users' intentions to continue using virtual interactive platforms. The results showed a positive and significant relationship between user satisfaction and all four motivation types, consistent with previous studies by Rahi et al. [32,33,54] and Lin et al. [16]. This suggests that higher user satisfaction with virtual interactive platforms leads to greater motivation to continue using them.

Finally, we explored the associations between user satisfaction, motivation, and continued intention to use virtual interactive platforms. The findings revealed that all factors

were related to continued use intention, with identified motivation having the strongest impact ($\beta = 0.136$; $p < 0.000$), followed by external motivation ($\beta = 0.129$; $p < 0.01$). Intrinsic motivation had the lowest impact ($\beta = 0.073$, $p < 0.05$). This indicates that when employees perceive a particular virtual interactive platform as important, they are more likely to continue using it. In contrast, intrinsic motivation has less influence in predicting continued usage. In work-related meeting platforms in SMEs, employees' motivation is driven by specific goals and community regulations, rather than individual emotions. Even if the platform is not engaging, employees can still be autonomously motivated if they recognize its value or significance [79].

### 6.2. Theoretical Implications

This study contributes to the literature on virtual interactive platforms in several ways. Firstly, it confirms telepresence and interactivity as key elements of virtual meeting services that enhance user satisfaction and drive continued usage. By focusing on employees in SMEs, this study provides valuable insights into the behaviors and attitudes of remote workers with regard to virtual interactive platforms.

Second, this article proposes the potential contribution of the OIT theory in enhancing our understanding of the intricate factors influencing users' continuous intention to use virtual interactive platforms within the hybrid workplace. OIT is a theoretical framework used to explain the various types of extrinsic motivation in different social contexts. It was previously investigated in consumers' passion for brands by Gilal et al. [80] and in determining knowledge-creation behaviors by Ma et al. [81]. Implementation of OIT will pave the way for future research. This study has provided an illustration of how the four motivation regulations of OIT theory are integrated effectively with other elements, including technology features (telepresence and interactivity), user satisfaction, and continued use intention. This is the first paper to investigate OIT theory and these factors together.

Thirdly, while previous theories such as the expectation-confirmation model (ECM), technology acceptance model (TAM), and unified theory of acceptance and use of technology (UTAUT) have confirmed the significant influence of user satisfaction on continued usage intention, few studies have explored the mediating effects of extrinsic motivation between satisfaction and intention. Our findings highlight that higher user satisfaction increases extrinsic motivation, thus encouraging the continued use of virtual interactive platforms in telework. This study makes a significant contribution to the theoretical and behavioral literature, particularly in the domain of virtual conference technology.

### 6.3. Practical Implications

The findings of this study hold significant practical implications for both firm managers and platform providers.

For firm managers in organizations adopting hybrid work models, the use of virtual interactive tools is crucial for fostering collaboration [10], enhancing productivity [9], and mitigating environmental impact [19,20]. Managers should be aware that employees are motivated to use specific services due to individual purposes (identified motivation) and social pressures (external motivation) such as managerial requirements, company regulations, or colleagues' preferences. Therefore, sustainability considerations should be integrated into decision-making processes regarding hybrid work policies and technology investments. This integration maximizes the sustainability benefits of virtual interactive platforms in hybrid work settings. While technical characteristics do not directly impact continued usage intention, they do have a positive direct effect on user satisfaction. This indicates that user satisfaction is likely to vary with changes in technical quality. Managers can effectively improve employees' satisfaction with a specific virtual interactive platform through training sessions, hands-on support, or comprehensive resources such as tutorials, videos, or user guides to familiarize employees with the platform.

For platform providers, it is crucial to understand the needs of workplace communities and develop services that meet specific user demands, thereby enhancing their experiences.

For example, educational institutions tend to choose platforms that provide a wide range of instructional resources and facilitate connection with students, while for-profit enterprises prefer platforms that offer visual benefits and user-friendly settings. Providers should also pay close attention to technical issues, since user satisfaction significantly influences user motivation and continued usage intention. The design of virtual interactive platform systems should aim to facilitate immersive user experiences, ensuring appropriate visual and auditory support for each activity and enabling users to have active control during work-related interactions. Active control requires the system to be engineered in a way that allows users to comfortably operate and participate in the entire meeting. This ensures a smooth and engaging communication experience without technical difficulties, regardless of the environment.

### *6.4. Limitations and Future Works*

This study has addressed important theoretical and empirical considerations regarding users' intentions to continue using virtual interactive platforms. However, certain limitations should be acknowledged. Firstly, the sample consisted of employees from selected organizations in Daegu, thus limiting the generalizability of the findings to the wider population of virtual interactive platform users in South Korea. To enhance generalizability, future studies should replicate the model in diverse regions.

Secondly, the study focused only on two technical dimensions of the platform systems, as they are the prominent characteristics of virtual services, both suggested by Tran [21] and supported by previous research [22,23,48,49,77]. Given recent rapid technological advancements, it is necessary to comprehensively examine the technological aspects, considering that future users may expect platforms to support interaction with virtual content in multifaceted spaces, such as 3D environments or the Metaverse. Further research should explore additional factors to provide a broader understanding of the technology-based service industry.

Thirdly, data collection and analysis were completed based on a survey questionnaire and empirical research to avoid several biases and enhance the research validity. To enhance the comprehensiveness of future research, however, it is recommended to supplement quantitative approaches with qualitative methods such as interviews and observations. This combined approach will provide deeper insights into participants' actual behaviors, and facilitate a more detailed understanding of both theoretical and managerial implications.

## 7. Conclusions

The appearance of virtual interactive platforms has facilitated hybrid work styles, offering potential contributions to sustainability goals. Hybrid work can reduce carbon emissions, conserve resources, foster partnerships, and enable global collaboration without extensive travel or physical office spaces [82]. However, the rapid adoption of virtual technologies is not without risks. Previous studies have highlighted the challenges associated with limited social and physical awareness and uninteresting meeting experiences in hybrid work [83]. Dissatisfaction with virtual interactive systems can lead to poor performance, disengagement, and loss of control during meetings [84]. Therefore, an in-depth understanding of individual motivations and usage intentions toward virtual interactive platforms is crucial for enhancing the effectiveness of hybrid work and promoting sustainable development. This study provides valuable insights for managers and platform providers, guiding adoption strategies, training provision, behavioral change, feedback gathering, and the monitoring of these platforms' impact on sustainable practices.

**Author Contributions:** H.T.D.T. and M.K. contributed to the design and implementation of the research, to the analysis of the results, and to the writing of the manuscript. All authors have read and agreed to the published version of the manuscript.

**Funding:** This research received no external funding.

**Institutional Review Board Statement:** Ethical review and approval were waived for this study due to the utilization of survey research methods, which inherently present minimal risk to participants, ensure voluntary participation, maintain confidentiality, and adhere to ethical guidelines.

**Informed Consent Statement:** Informed consent was obtained from all subjects involved in the study.

**Data Availability Statement:** The data that support the findings of this study are available from the corresponding author, Kim, M., upon reasonable request.

**Acknowledgments:** I am deeply grateful to my supervisor, Kim, M., for her unwavering support and guidance during this research. Her expertise and encouragement were invaluable. I also thank the participants for their willingness to take part in the study, without whom this research would not have been possible. All individuals mentioned have given their consent to be acknowledged.

**Conflicts of Interest:** The authors declare no conflict of interest.

## Appendix A

**Table A1.** Factor Loadings Output.

| Variables | 1 | 2 | 3 | 4 | 5 | 6 | 7 | 8 |
|---|---|---|---|---|---|---|---|---|
| TEL1 | **0.794** | 0.200 | 0.159 | 0.103 | 0.089 | 0.148 | 0.064 | 0.157 |
| TEL4 | **0.781** | 0.081 | 0.063 | 0.155 | 0.111 | 0.157 | 0.075 | 0.151 |
| TEL2 | **0.776** | 0.116 | 0.128 | 0.080 | 0.003 | 0.094 | 0.131 | 0.067 |
| TEL3 | **0.754** | 0.146 | 0.101 | 0.138 | 0.087 | 0.107 | 0.052 | 0.200 |
| INT1 | 0.162 | **0.768** | 0.137 | 0.097 | 0.091 | 0.137 | 0.122 | 0.111 |
| INT3 | 0.056 | **0.766** | 0.123 | 0.120 | −0.062 | 0.089 | 0.091 | 0.049 |
| INT4 | 0.136 | **0.760** | 0.095 | 0.152 | 0.114 | 0.034 | 0.079 | 0.204 |
| INT2 | 0.192 | **0.744** | 0.193 | 0.097 | 0.114 | 0.102 | 0.039 | 0.155 |
| SAF2 | 0.080 | 0.171 | **0.787** | 0.182 | 0.146 | 0.093 | 0.128 | 0.041 |
| SAF3 | 0.134 | 0.156 | **0.756** | 0.171 | 0.130 | 0.083 | 0.109 | 0.001 |
| SAF4 | 0.145 | 0.031 | **0.734** | 0.195 | 0.198 | −0.035 | 0.210 | 0.085 |
| SAF1 | 0.107 | 0.220 | **0.694** | 0.107 | 0.152 | 0.155 | −0.002 | 0.118 |
| CUI3 | 0.178 | 0.150 | 0.123 | **0.762** | 0.141 | −0.001 | 0.152 | 0.081 |
| CUI2 | 0.181 | 0.153 | 0.274 | **0.725** | 0.118 | 0.189 | 0.168 | 0.113 |
| CUI1 | 0.179 | 0.076 | 0.253 | **0.723** | 0.128 | 0.226 | 0.216 | 0.182 |
| CUI4 | 0.041 | 0.218 | 0.181 | **0.601** | 0.231 | 0.143 | 0.126 | 0.194 |
| EXT1 | 0.061 | 0.069 | 0.156 | 0.162 | **0.852** | 0.063 | 0.042 | 0.056 |
| EXT3 | 0.083 | 0.068 | 0.133 | 0.163 | **0.833** | 0.114 | 0.044 | 0.025 |
| EXT2 | 0.098 | 0.061 | 0.240 | 0.095 | **0.831** | −0.016 | 0.006 | 0.087 |
| INJ2 | 0.148 | 0.081 | 0.106 | 0.074 | 0.065 | **0.830** | 0.172 | 0.117 |
| INJ3 | 0.117 | 0.112 | 0.091 | 0.150 | 0.044 | **0.812** | 0.098 | 0.132 |
| INJ1 | 0.202 | 0.143 | 0.059 | 0.135 | 0.061 | **0.792** | 0.169 | 0.098 |
| IDE2 | 0.065 | 0.100 | 0.096 | 0.174 | 0.043 | 0.167 | **0.834** | 0.149 |
| IDE3 | 0.103 | 0.087 | 0.121 | 0.146 | 0.083 | 0.123 | **0.832** | 0.116 |
| IDE1 | 0.142 | 0.138 | 0.182 | 0.185 | −0.035 | 0.172 | **0.763** | 0.164 |
| INTR1 | 0.158 | 0.181 | 0.072 | 0.139 | 0.031 | 0.164 | 0.151 | **0.818** |
| INTR3 | 0.199 | 0.169 | 0.039 | 0.125 | 0.024 | 0.059 | 0.174 | **0.780** |
| INTR2 | 0.194 | 0.141 | 0.102 | 0.153 | 0.133 | 0.154 | 0.108 | **0.751** |

Notes: Bolded coefficients are the factor loadings.

**Appendix B**

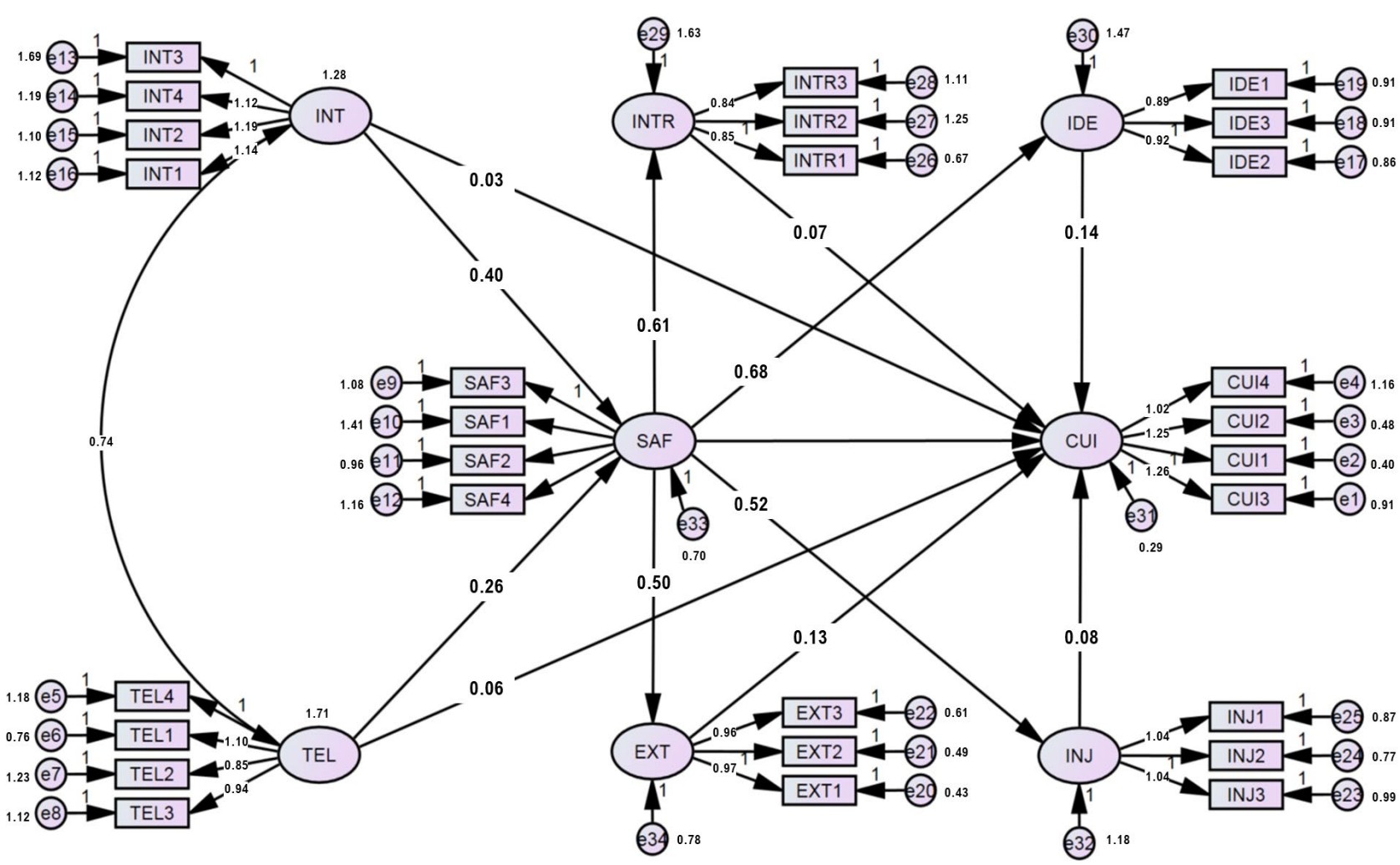

**Figure A1.** The Research Model and Coefficients (SEM).

**Appendix C**

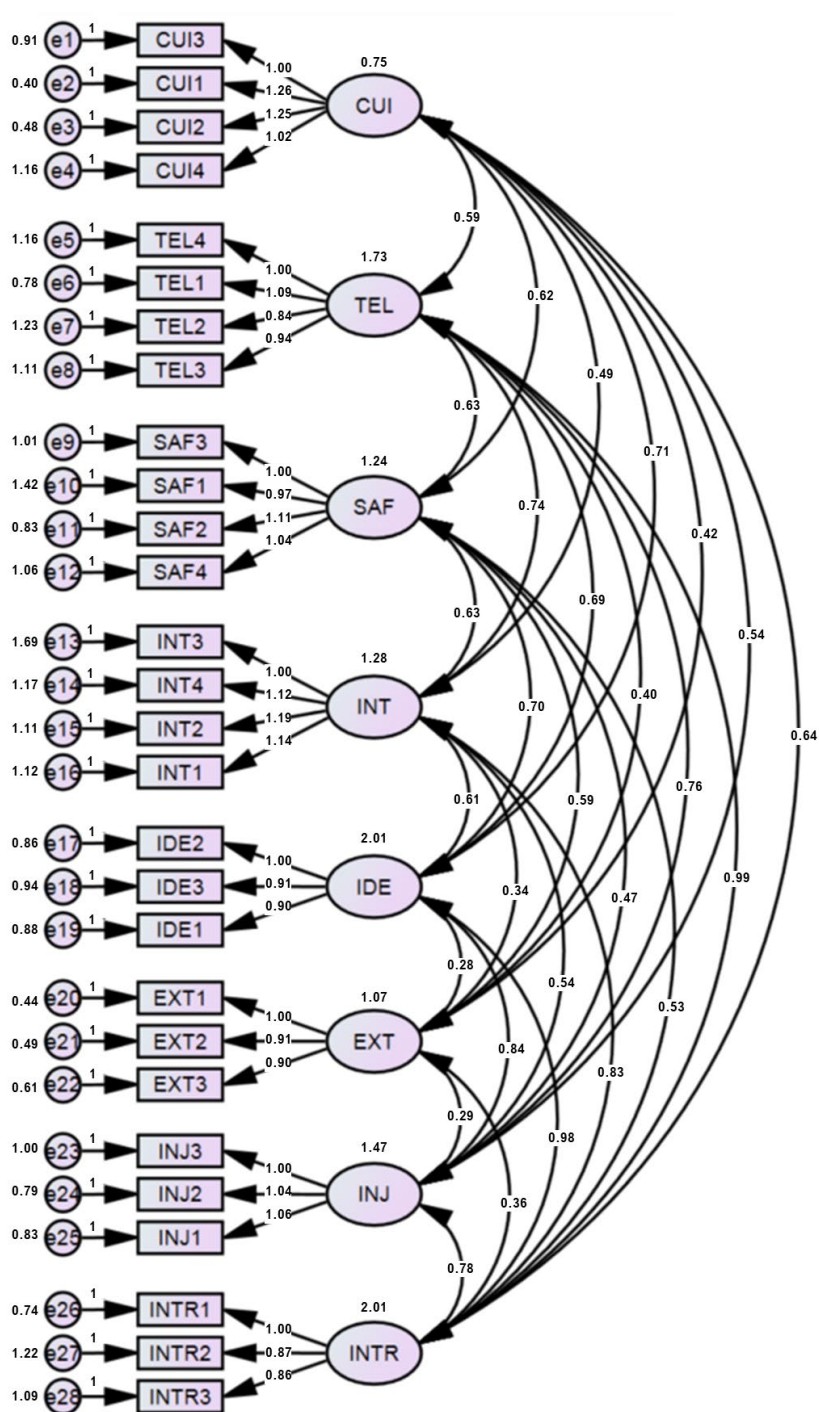

**Figure A2.** The Research Model and Coefficients (CFA).

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
