# Peer review of "Factors Influencing the Continued Intent to Use Virtual Interactive Platforms in Korean Small- and Medium-Sized Enterprises for Remote and Hybrid Work"

_sustainability, doi:10.3390/su15139972_

Round 1
Reviewer 1 Report
The topic raised by the authors is extremely important and current in the post-pandemic world.
The conclusions may be useful not only from the point of view of the Republic of Korea, but also other markets.
Is the use of the acronym VIP completely justified? It creates a comprehension problem for the reader. Are you able to change it?
Is the collected sample sufficient? Please also state the opposite position.
When was the study conducted? How long did it take? What specific research method was used? Please describe it.
Author Response
Dear Reviewer,
Thank you for dedicating your time and offering valuable insights into my manuscript. Your feedback is greatly appreciated, and I would like to express my sincere gratitude for your careful consideration.
I have incorporated all suggested changes into the new version of the manuscript. These revisions are clearly marked in RED in the manuscript to ensure visibility. In response to your comments, please find the following key points addressed:
Point 1: Is the use of the acronym VIP completely justified? It creates a comprehension problem for the reader. Are you able to change it?.
Response 1: The use of “VIP” has been revised throughout the text to avoid comprehension problems for the reader based on your recommendation.
Point 2: Is the collected sample sufficient? Please also state the opposite position.
Response 2: Please review more details in “4.2. Methods setting and Sample” on page 9 of the manuscript.
- The number of responses was regarded as optimal if it was greater than 200, as proposed by Thompson (2004). It is also essential to a measurement model’s aspects to calculate the sample size for decreasing the errors of sampling since typical sample size calculation rules (e.g., power, effect size, or minimal sample size) ignore the study model’s specifications (Wolf et al., 2013).
- This study calculated the minimum sample size to determine the suitable size for model structure and detect the effect through the “A-priori Sample Size Calculator for Structural Equation Models” introduced by Dr. Daniel Soper. Specifically, we determined the expected effect size to be 0.3 (medium), the desired statistical power level to be 0.8, the number of latent variables to be 8, the number of observable variables to be 28, and the probability level to be 0.05 (Dattalo, 2009) and got a result of the minimum recommended sample size: N=177. As the sample size (N=353) was further than 177, we can conclude that the results obtained from such a dataset may enhance its validity and reliability and increase the generalizability of the theoretical model.
Point 3: When was the study conducted? How long did it take? What specific research method was used? Please describe it.
Response 3: To avoid bias in sampling techniques, this study implemented two independent data collection approaches during the period from 20 January to 28 February 2022 (within 40 days). Please review more details in “4.1. Participants and Procedure” on page 8 of the manuscript.
This study used literature and empirical research methods to achieve the mentioned purposes.
- Regarding a conceptual approach, this study checked the factors and measures of individual intention, motivation, satisfaction, and features of virtual technology-based platforms from prior studies [8, 10, 15, etc.].
- In terms of an empirical approach, we analyzed the collected data using structural equation modeling (SEM), which followed the two-step process recommended by Hair et al. (2010). More specifically, Exploratory Factor Analysis (EFA) was performed using the IBM SPSS 25.0 program to examine the structure of the predictor variables. We then used SPSS Amos 26.0 software to run a Confirmatory Factor Analysis (CFA) on the measurement model and test constructed hypotheses using structural path analysis.
Once again, thank you for your valuable input and feedback. I eagerly await your review of the revised manuscript. Your time, expertise, and support are greatly appreciated.
I look forward to the possibility of publishing our work in the Sustainability journal.
Kind regards,
Hang Tran.

Reviewer 2 Report
Dear Authors,
I have read with a great deal of interest your manuscript, since the subject has such a degree of novelty and great applicability. However, in order to improve your research, here are some recommendations that you might follow:
- please restructure the title according to your findings;
- please provide a literature substantiation for each of the hypotheses that you intent to test;
In regard to the measures development, please provide literature which describes all the variables used and please validate the format that you used.
- according to general methodological agreements, you cannot analyze online and on site provided questionnaires altogether, due to bias exposure; what did you do as to avoid common bias in this regard?
Have you respected the GDPR norms for both online and on site provided questionnaires? Were the respondents remunerated? Please provide data with literature support for your choices.
- how long was the data collection period? Please provide data; when were these steps undertaken? Please provide the time period.
- please provide data in regard to the software used and the version available;
- did you run a pilot study as the methodological approaches require? Please provide data;
- please restructure the Materials and Methods chapter as following:
1 participants and procedure; 2. Methods setting and sample; 3. measures; 4. Research methodology followed by the Results chapter.
- within the Results chapter, please support with previous literature findings your results;
- please cite all the indices ranges that you used, since literature does or not always agrees on the same intervals;
- the statistical analysis is sufficient as for (in)confirming your hypothesis? Please provide data with literature support for your choices.
- please provide the model and its coefficients resulted from your statistical analysis;
- please provide data in regard to your factor loadings on the output model;
- results should be discussed in the light of previous research;
- please provide a title that regards hypothesis development, testing and results, of course, in the light of previous research; it would also be interesting to see within this chapter that you analyze your hypothesis groups by following the logic presented within literature review;
- theoretical and practical implications need to be further enhanced; similar to conclusions and limitations.
- the References chapter needs to be greatly improved
As for literature support in regard to your model, you might find useful to read:
Hair Jr, J. F., Howard, M. C., & Nitzl, C. (2020). Assessing measurement model quality in PLS-SEM using confirmatory composite analysis. Journal of Business Research, 109, 101-110.
Sarstedt, M., Radomir, L., Moisescu, O. I., & Ringle, C. M. (2022). Latent class analysis in PLS-SEM: A review and recommendations for future applications. Journal of Business Research, 138, 398-407.
McIntosh, C. N., Edwards, J. R., & Antonakis, J. (2014). Reflections on partial least squares path modeling. Organizational Research Methods, 17(2), 210-251.
Best regards,
English language is coherent and adequate
Author Response
Dear Reviewer,
Thank you for dedicating your time and offering valuable insights into my manuscript. Your feedback is greatly appreciated, and I would like to express my sincere gratitude for your careful consideration.
I have incorporated all suggested changes into the new version of the manuscript. These revisions are clearly marked in RED both in the manuscript and in the cover letter response to your recommendations to ensure visibility.
Please see the attachment for more details.
Kind regards,
Hang Tran.

Reviewer 3 Report
Review Report Manuscript ID: sustainability-2445393
Type of manuscript: Article
Title: The continued use intention of users toward virtual interactive platforms in a sustainable hybrid work model
Date: 03 June 2023
Dear Authors,
This study analyses the “The continued use intention of users toward virtual interactive platforms in a sustainable hybrid work model”. The topic is of high interest, especially after the accelerating virtual working by COVID-19 in these years. The SEM models are appropriate method to the analysis with relevant hypothesis on the relationship between variables. Results derive logically from the results obtained. The presentation is precise and clear. The data, the analysis and results and discussion sections are all well-structured. I think this work could be published. In addition, I think the study and manuscript fits the journal scope and provides new information.
Kindly consider the below comments and suggestions. The below comments and suggestions are meant to improve the quality of your manuscript. The publication of this articles in sustainability subjects to minor modifications and clarifications.
1. The last three lines suddenly appear enhancing the use experiences. However, the reasoning of enhancing experiences should be based on the hypothesis or the experiment as data collected. The authors are suggested to be addressed this issue. Does it additional finding that revealed during the survey is implementing?
2. The figures are too small. I suggest the authors to enlarge them for the figure of the hypothesis (figure 1), and that in the hypotheses result (figure 2).
3. The abbreviation terms are suddenly appears in the result, such as CFA and EFA. The methodology needs to be completely introduced in advance to the results are reported, and the full name of the terminology should be offer as well.
4. Please recheck if the notes below table 2, 3 and 5 are misplaced.
Author Response
Dear Reviewer,
Thank you for dedicating your time and offering valuable insights into my manuscript. Your feedback is greatly appreciated, and I would like to express my sincere gratitude for your careful consideration.
I have incorporated all suggested changes into the new version of the manuscript. These revisions are clearly marked in RED in the manuscript to ensure visibility. In response to your comments, please find the following key points addressed:
Point 1: The last three lines suddenly appear enhancing the use experiences. However, the reasoning of enhancing experiences should be based on the hypothesis or the experiment as data collected. The authors are suggested to be addressed this issue. Does it additional finding that revealed during the survey is implementing?
Response 1: Regarding this point, the full explanation of findings and extra implications were greatly improved. Please review more details in “6. Discussions” on pages 15~17, and “7. Conclusion” on page 17 of the manuscript.
Point 2: The figures are too small. I suggest the authors to enlarge them for the figure of the hypothesis (figure 1), and that in the hypotheses result (figure 2).
Response 2: All illustrative figures have been enlarged and more details have been added based on your recommendation. Please review them on pages 7, 14, 18 and 19 of the manuscript.
Point 3: The abbreviation terms are suddenly appears in the result, such as CFA and EFA. The methodology needs to be completely introduced in advance to the results are reported, and the full name of the terminology should be offer as well.
Response 3: The mistakes related to abbreviation terms have been fixed. Furthermore, we have attempted to describe in more detail the modules used, calculated coefficients, and software version applied. We have explained and added more information related to the validation of the instrument with the research data. Please review more details in “4.4. Research Methodology” on page 11, “5. Results” on pages 11~14, and Appendix A and B on pages 18~19 of the manuscript.
Point 4: Please recheck if the notes below table 2, 3 and 5 are misplaced.
Response 4: All tables and their notes have been checked based on your recommendation.
Once again, thank you for your valuable input and feedback. I eagerly await your review of the revised manuscript. Your time, expertise, and support are greatly appreciated.
I look forward to the possibility of publishing our work in the Sustainability journal.
Kind regards,
Hang Tran.

Reviewer 4 Report
The authors of the paper titled "The continued use intention of users toward virtual interactive platforms in a sustainable hybrid work model" employs an empirical approach to identify the features of VIPs that strengthen user intention to continue using the platform service and develop an integrative virtual technology usage model based on the concepts of technology features, motivation theory, user satisfaction, and continued use intention of employees in SMEs.
This topic is interesting and has potential but in my opinion, it should be at least partially developed before publication with further work described in the abstract and summary. I'm also not sure if Sustainability is best appropriate for this article to be published. I suggest changing the Journal to another MDPI Journal.
For the future, I have also some editorial comments.
1. The text formatting should be corrected, e.g. tables divided into two pages, highlighting the text with color (in chapter 4.1), chapter names cannot be left at the end of the page (e.g. chapter 2.3), and other editorial errors.
2. If we enumerate hypotheses after a colon, then each hypothesis cannot be followed by a dot ending a sentence. The dot should be at the end, and hypotheses should be separated by a comma or semicolon.
3. Some graphics should be clearer (font in Figure 2).
Author Response
Dear Reviewer,
Thank you for dedicating your time and effort to reviewing our paper. We highly appreciate your valuable feedback and have carefully considered your comments regarding the suitability of our paper for the Sustainability journal.
In response to your concerns, we have made significant improvements to the entire article by implementing the following main points.
- We have reconstructed the entire paper, providing additional explanations for each section to enhance clarity and coherence.
- We have expanded the literature background for all sections included in the article, ensuring a comprehensive understanding of the research context.
- To provide more information and coefficients, we have included relevant figures and tables, enabling readers to gain a deeper insight into our findings.
- We have given significant attention to developing the References section, ensuring a comprehensive and up-to-date list of relevant sources.
Furthermore, we would like to emphasize why we believe our study is appropriate for the Sustainability journal.
- Previous studies have acknowledged the impact of the efficient use of virtual interactive services on the sustainability of hybrid work arrangements. However, few studies have specifically focused on identifying predictive factors for the continued usage of virtual interactive platforms among SME employees.
- Understanding individual motivation and usage intentions for virtual interactive platforms is crucial for enhancing hybrid work and promoting sustainable development. Dissatisfaction with the virtual interactive system has been associated with detrimental outcomes such as poor performance, disengagement, and loss of control during meetings.
- Our study provides a robust literature background that offers managers in organizations, as well as platform providers valuable insights. This background aids in driving adoption, providing relevant training, fostering behavioral change, gathering feedback for improvement, and monitoring the impact of these platforms on sustainable practices.
- The theoretical and practical implications of our study extend to managers in organizations seeking to maximize the sustainability benefits of virtual communication tools in hybrid work environments. Additionally, our findings provide meaningful information for platform developers to enhance their services.
In response to your comments regarding the editing style, please see page 2 of the attachment for more details.
Kind regards,
Hang Tran.

Round 2
Reviewer 2 Report
Dear Authors,
I congratulate you for your timely manner of answering the suggestions from the first review round.
Best regards,
Minor spelling errors. The manuscript is well written, with an use of academic language well suited for publishing further.
Reviewer 4 Report
After reviewing the authors' responses and the changes to the manuscript, I believe that the article can be accepted in its present form.